# Sustainable Transportation Infrastructures in Iowa—Goals and Practices

**Hosin Lee** [1,*], **Byungkyu Moon** [2] and **Jeongbeom Lee** [1]

[1] Iowa Technology Institute, Department of Civil and Environmental Engineering, The University of Iowa, Iowa City, IA 52242, USA; jeongbeom-lee@uiowa.edu

[2] Applied Research Associates, Inc., 100 Trade Centre Drive, Suite 200, Champaign, IL 61820-7233, USA; bymoon@ara.com

[*] Correspondence: hosin-lee@uiowa.edu

**Abstract:** The need to incorporate sustainability principles and practices is increasing for environmental and economic reasons. It is imperative to identify and operationalize sustainability strategies into core administrative, planning, design, construction, operational, and maintenance activities for the transportation infrastructure systems by integrating sustainability into decision-making processes. The primary goal of this study is to develop an implementation plan for achieving more sustainable transportation infrastructure systems in Iowa. This research aims to guide the adoption of sustainable strategies, balancing cost, performance, and environmental impact in transportation infrastructure development. This paper presents efforts to develop a methodology for identifying the best sustainable practices for implementation in transportation infrastructure practices in Iowa by surveying state DOTs to learn about their sustainability goals and practices, identifying existing sustainability attributes and sustainable practices, and developing a GIS database where construction, materials and performance data of sustainable practices can be stored and analyzed.

**Keywords:** sustainability; survey of state DOTs; Python program on ArcGIS database; sustainable practices

## 1. Introduction

Recent sustainability initiatives emphasize environmental stewardship and ecological preservation at national and local levels. Sustainability is a concept that has been introduced previously in the design and construction of transportation projects. For example, the need to incorporate recycled materials and locally available materials in pavements has long been emphasized from both economic and environmental viewpoints. Nationally, the average amount of reclaimed asphalt pavement (RAP) in new asphalt pavement structural layers during 2020 was 21.9 percent, which is the highest level reported since the survey began in 2009 [1]. However, the NAPA survey found that the most common factor limiting the utilization of RAP was "low specification limits", i.e., a 20% limit of RAP in a surface layer. Because Iowa DOT and local public agencies are building, operating, and maintaining their transportation infrastructure systems with constrained budgets, there are critical needs to maximize service and performance quality while controlling costs and environmental impact, integrate sustainability in decision-making processes for planning, designing, building, and managing transportation infrastructure systems, and incorporate sustainability principles and practices in their everyday operations.

Milani et al. [2] provide a comprehensive framework for sustainable transport infrastructure and discuss the importance of flexibility in transport infrastructure, especially in light of unpredictable shifts in demand, such as those experienced during the COVID-19 pandemic. The need to incorporate sustainability principles and practices is increasing due to both environmental and economic reasons. It is critical to identify and operationalize sustainability strategies into core administrative, planning, design, construction, operational,

and maintenance activities for the transportation infrastructure systems using a comprehensive systems approach and the integration of sustainability in decision-making processes.

The primary goal of this study is to develop an implementation plan for achieving more sustainable transportation infrastructure systems for Iowa DOT and local road agencies. This research aims to develop a methodology for identifying the best sustainable practices for implementation in transportation infrastructure practices in Iowa by developing a shelf-ready idea based on identified target areas to be put into practice through small-scale pilot projects.

This paper discusses efforts to identify sustainable goals and practices by surveying 50 state Departments of Transportation (DOTs). It presents a comprehensive study on integrating sustainability in Iowa's transportation infrastructure by developing the database of sustainable practices with implementation records (DOSPIR), which can serve as a central repository of construction, materials, and performance data of sustainable practices, facilitating easy access and analysis of its effectiveness for transportation engineers.

## 2. Background

Sustainability is a concept that considers a long-term view of projects, considering costs and benefits over a lifetime rather than concentrating on a short-term cost. However, unlike buildings that adopted many sustainable practices through the LEED (leadership in energy and environmental design), transportation infrastructure has been largely neglected regarding sustainability. Transportation infrastructure is essential to sustainability for both urban and rural communities in the U.S. Therefore, the need to incorporate sustainability principles and practices in constructing and maintaining transportation infrastructure is proliferating due to both environmental and economic reasons.

Mead [3] provides a global perspective on sustainable transportation efforts and their impact on urban development. It discusses the progress and challenges in achieving sustainable transportation by promoting investments in walking and cycling infrastructure and various initiatives for creating more walkable and bike-friendly urban environments. Recently, a special issue was published on sustainable transportation infrastructure with a comprehensive view of the most recent research and advancements in sustainable transportation infrastructure, which covers a wide range of topics, including pedestrian and bicycle safety, complete streets, connected and automated vehicles, electric vehicles, and the impact of transportation infrastructure on urban heat and community severance [4].

The Federal Highway Administration (FHWA) is actively promoting sustainability through the INVEST (infrastructure voluntary evaluation sustainability tool) rating system, which was developed to encourage efforts towards sustainability in transportation projects [5]. As shown in Figure 1, FHWA supports activities to facilitate balanced decision-making among environmental, economic, and social values–the triple bottom line of sustainability [6]. As shown in Figure 2, 71 agencies adopted INVEST in their projects. Over 2000 users are registered for the INVEST with over 2400 projects.

Several sustainability rating systems have been developed in the past to encourage public agencies to adopt more sustainable practices. Agencies can use a sustainability rating system to evaluate the sustainability of their projects with a set of sustainability criteria. Each sustainability rating system differs in how it evaluates sustainable practices based on various criteria, with a different weight given to each criterion. The most common sustainability rating systems have been evaluated to identify how credits are distributed across three sustainability categories: economy, environment, and society. As shown in Table 1, we reorganized the original table by Simpson et al. [7] with respect to application areas, no. of criteria, and main features. As can be seen from Table 1, all rating systems are only applicable to highway projects except "Envision", which applies to different types of infrastructure projects. Particularly, the "Green Guide for Road" rating system includes the industry's best practices, which were adopted into the LEED program.

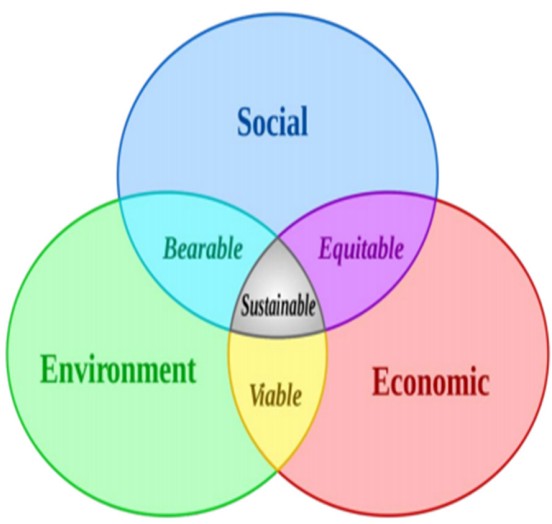

**Figure 1.** Triple bottom line of sustainability [6].

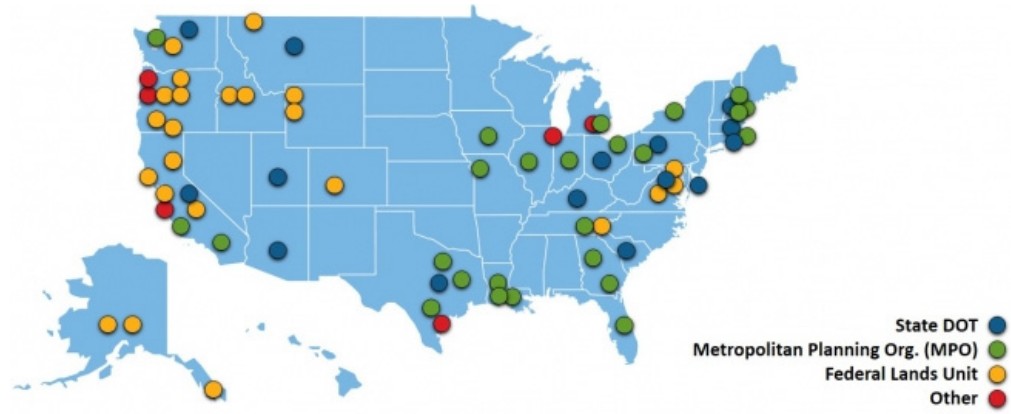

**Figure 2.** Users of infrastructure voluntary evaluation sustainability tool [5].

**Table 1.** Comparison of the ten most common sustainability rating systems.

| System | Developer | Applicability | Criteria | Main Features |
|---|---|---|---|---|
| BE2ST-in-Highways | University of Wisconsin-Madison | Planning/design | 7 categories, 10 criteria | Recycling, weights life-cycle assessment |
| Envision | ISI, Harvard U. | Planning/design constr./maint. | 5 categories 60 credits | Self-assessment, any point in the life cycle |
| Green Guide for Road | Stantec | Planning/design | 7 categories 35 credits | Industry's best practices, into LEED |
| Greenlites | New York State DOT | Planning/design constr./maint. | 5 categories 175 credits | Performance, identify areas of improvement |
| GreenPave | Ontario Ministry of Transportation | Planning/design constr./maint. | 4 categories 36 points | Focus on pavements |
| Greenroads | CH2M HILL/ U. of Washington | Planning/design construction | 6 categories 31 criteria | Roadway sustainability quantitative method |
| I-LAST | Illinois DOT, ACEC, IRTBA | Planning/design construction | 8 categories 153 criteria | Design and future construction phase |
| INVEST | CH2M Hill/FHWA | Planning/design constr./maint. | 68 criteria | Predefined/custom scorecards |
| CEEQUAL | Institution of Civil Engineers (UK) | Planning/design constr./maint. | 9 categories | Applicable to a wide range of project types |
| STARTS | N. American Sustain. Transp. Council | Planning/design constr./maint. | 6 categories 29 credits | Transportation and land use strategies |

The 2019 Sustainability Report by Minnesota DOT lists 38 metrics in 7 areas of transportation (10 metrics for transportation modes), facilities (9 metrics related to energy and water), fleet (6 metrics of alternative fuels), highway operations (5 metrics for lighting and deicing), roadside management (2 metrics of native plantings and snow fences), construction (3 metrics for sustainable pavements and recycling) and climate resilience (3 metrics related to culverts and precipitation) [8]. These 38 metrics are quite similar to 27 metrics developed by Iowa DOT's sustainability working group [9].

In Iowa, significant efforts have been made to integrate social, economic, and natural resource needs while creating a community forum for discussing and sharing ideas for a more sustainable tomorrow for Greater Des Moines [10]. The Iowa Economic Development Authority (IEDA) developed Iowa Green Streets Criteria to encourage sustainable community practices of (1) integrative design, (2) location + neighborhood fabric, (3) site improvements, (4) water conservation, (5) energy efficiency, (6) materials, (7) healthy living environment, and (8) operations, maintenance, and occupant engagement [11].

Iowa DOT has always been at the forefront of sustainable project development and design practices. Iowa DOT's project development process encompasses a design concept following a context-sensitive solutions (CSS) approach that fits the transportation infrastructure into the environment rather than altering a sensitive environment to fit the infrastructure [12]. The Bridges and Structures Bureau follows established Iowa DOT guidelines in using sustainable practices with the following concerns: (1) sedimentation and erosion control, (2) disturbance to wetland and farmland, (3) fastest track construction, (4) impact of footings and piers in surrounding environment, and (5) one new bridge to replace several old and smaller bridges [13].

## 3. Sustainability Survey

To identify their sustainability goals and practices, we have surveyed state DOTs that have or are currently implementing sustainability into their activities.

### 3.1. Sustainability Survey by AASHTO

In 2016, AASHTO created the Committee on Environment & Sustainability (CES), which reflected the growing importance and consideration of sustainability among AASHTO members. In 2020, CES established the Sustainability Working Group (SWG) to address climate change and reduce energy use, water use, and $CO_2$ emissions. The SWG conducted a survey of 50 state DOTs and received responses from 44 states [14].

Based on the survey, only 10 state DOTs have adopted a definition of "sustainability", and an additional 10 state DOTs consider sustainability as a factor in their strategic and project planning. Most state DOTs house sustainability initiatives and staff as part of existing offices except five state DOTs, which maintain a standalone sustainability office. Sixteen state DOTs have directives from governors or legislatures to address sustainability and implement formal sustainability programs. Most state DOTs are or plan to conduct the following four sustainability practices: (1) reducing facilities' energy use, (2) increasing infrastructure resilience, (3) increasing electric vehicle infrastructure, and (4) considering climate change when planning and designing projects. Many state DOTs provide direct general funds and indirect funds diverted from other programs, grants, and pilot programs in support of sustainability programs.

### 3.2. Sustainability Survey by the University of Iowa

Two surveys were conducted by the University of Iowa, and out of 50 state DOTs, we received 9 and 16 responses from each of the two surveys. The first survey asked the following three questions: sustainability goals, successful sustainable practices, and means to incorporate sustainability into the decision-making process. The second survey asked in the following three categories: sustainability evaluation metrics, incentives for sustainability implementation, and lessons learned from sustainability implementation. The survey results are summarized below.

### 3.2.1. First Survey on Sustainability Goals, Practices, and Means to Incorporate Sustainability

Based on the AASHTO survey results, twenty-five state DOTs were selected for our follow-up survey. We have received nine responses, and overall, most state DOTs were interested in sustainability, but a small number of state DOTs have adopted the sustainability concept. Based on the nine responses, the top five agency-wide sustainability goals are reuse and recycling, saving energy, increasing life cycle, minimizing waste, and clean air and water.

The most common successful sustainable practices are recycling asphalt and concrete pavements, installing LED lights, planting native species, utilizing recycled materials, keeping a database of sustainable practices, and converting traffic signals to roundabouts. The most popular means to incorporate sustainability into the decision-making process are modifying specifications to allow sustainable practices, reaching out to communities and stakeholders, creating a sustainability working group/department, measuring economic/social/environmental effects, and prioritizing sustainable strategies.

### 3.2.2. Second Survey on Sustainability Evaluation Metrics, Lessons Learned and Incentives

For the second round of the survey, surveys were sent to all fifty State DOTs, and we received 16 responses. Regarding sustainability evaluation metrics, only 3 states replied that they have such metrics, whereas the remaining 13 states did not. Regarding the lessons learned from sustainable practices, 8 states implemented the sustainable practices with lessons learned such as engagement with key staff members, reaching out to other agencies, support from leadership, adequate funding, not overthinking, not being afraid to make mistakes, need standard measurable metrics, incentivize contractors for more sustainable practices, and sharing best practices. Regarding providing incentives for implementing sustainable practices, no state is currently providing any incentive for sustainable practices.

## 4. Sustainable Practices

Suprayoga et al. [15] present a systematic approach to evaluating the sustainability of road infrastructure projects, which includes a detailed analysis of sustainability criteria to cluster and group various sustainability indicators. It provides a robust framework for implementing sustainability criteria in transportation infrastructure projects. It is challenging to identify sustainability attributes since experts differ in defining them, which are critical to different bureaus within the agency.

Based on evaluation results of five existing sustainability rating systems (Envision, GreenLITES, Greenroads, I-LAST, and INVEST), the highest weight was allocated to the environment category (16.4%) followed by materials category (14.2%), water quality/usage category (11.0%), and energy category (6.6%) [16]. Among 34 criteria adopted by these five sustainability rating systems, the most common criteria are recycled content/materials and locally provided/regional material in all five systems, followed by stormwater treatment and reduced electricity/energy consumption in four systems and habitat restoration, noise abatement, energy efficiency, stray light reduction, and pavement reuse in three systems.

A sustainable practice can be defined as a method to lower negative impacts on society and the environment. However, their relative impacts on society and the environment are difficult to quantify. Based on our experiences of conducting sustainable transportation infrastructure research, we developed Table 2, which summarizes 14 sustainable practices in construction and maintenance areas with their estimated impacts on cost, performance, sustainability attributes, and limitations. Under the cost and performance columns, "+", "=", and "−" symbols indicate if the sustainable practices increase, equal, and decrease cost and performance compared to the typical practices, respectively.

**Table 2.** Cost, performance, sustainability attributes and limitations of sustainable practices.

| Sustainable Practices | Cost | Performance | Sustainability Attributes | Limitations |
|---|---|---|---|---|
| HMA-RAP/RAS | − | = | Increased use of RAP/RAS Rutting Resistant | Increased cracking Cost of rejuvenator |
| Recycled concrete aggregates (RCA) | − | = | Conserves aggregates Freeze-thaw resistant | Dust/noise/wastewater lower RCA strength |
| UHPC concrete | + | + | Increased strength Smaller section | Cost of steel fibers and admixtures |
| Porous asphalt pavement | + | − | Noise reduction (−3.8 dB) Stormwater management | Clogging of pores |
| Pervious concrete pavement | + | − | Quiet (3–8% lower) Reduce splash/hydroplaning | Joint deterioration Debonding/local distress |
| HMA-ground tire rubber (GTR) | + | + | Perform better than PMA mix Abundant supply of GTR | Hard to work with Difficult to compact |
| Railcar bridge | − | = | Recycle railcars, conserves materials | Lack of Availability Less Durability |
| Cold/hot in-place recycling | − | =/+ | Saving asphalt and gravel, less transportation cost | Lack of performance Lack of quality control |
| Rubblization of concrete pavement | − | −/= | Less construction time | Lower strength Subgrade failure |
| Open graded friction course | = | = | Noise reduction (−4 dB) Cost-effective in rural areas | Not cost-effective, no Performance in cool climates |
| Diamond grinding of concrete pavement | − | + | Noise reduction Better ride quality/friction | Dust during grinding |
| Bio binder | − | −/= | Rutting and cracking-resistant Eco-friendly | Increase fatigue cracking |
| High-friction surf. treatment (HFST) | − | + | Reduces crashes and fatalities | Delamination from existing surface |
| Otta seal on gravel roads | + | + | Use of uncrushed aggregates Impermeable surfaces | Cont. rolling (8 weeks) No structural capacity |

## 5. Development of DOSPIR

Over the past decades, numerous test sections applying sustainable practices have been built throughout Iowa, and their construction and performance evaluation reports have been published. However, because the construction and performance evaluation results of these test sections have not been stored in an online computer database, they could not be easily searched by future users of the same sustainable practices.

Zhao et al. [17] proposed that the pavement management system (PMS) should incorporate sustainability principles and methods for assessing the environmental and economic impacts of using sustainable practices. While PMS is a comprehensive software tool that is effective for the overall maintenance and management of road infrastructure, its uniformly segmented database with no consideration of specific test sections makes it challenging to locate test sections constructed with sustainable practices.

Therefore, the database of sustainable practices with implementation records (DOSPIR) was developed to include sustainable practices along with their field performances. By providing a centralized platform for a transportation sustainability-focused database, DOSPIR would enable researchers to conduct comparative analyses across various geographical locations and over an extended timeframe. This would significantly enhance the ability to monitor, assess, and predict the long-term sustainability impacts of transportation projects.

DOSPIR was developed using the programming language Python on the ArcGIS Pro platform. ArcGIS Pro was used to store construction and performance monitoring

data and spatially join various datasets containing locations of test sections, performance data, and crash data. Python was used to develop a versatile and powerful data structure for handling and analyzing georeferenced data. This step involves refining the data by eliminating records with missing values and outlier data reduction and adding analyzed data. Such data processing and analysis steps are crucial for a comparative analysis of the impacts of implemented sustainable practices.

Python's data visualization tools are used to plot bar charts based on the spatially joined GIS database. To update them, users can download the latest locations of sustainable practices and performance/crash data files and run a DOSPIR with a new set of data files. By running the DOSPIR again with new datasets, the data sets will be automatically spatially joined, and visual representations will be updated with new sets of data files. Further, DOSPIR's online accessibility and spatial analysis functions would facilitate the ongoing data updates. Currently, as shown in Figure 3, a prototype DOSPIR includes four sustainable practices of asphalt pavement recycling, high-friction surface treatments, ultra-high-performance concrete (UHPC) bridges, and roundabouts, which are discussed below.

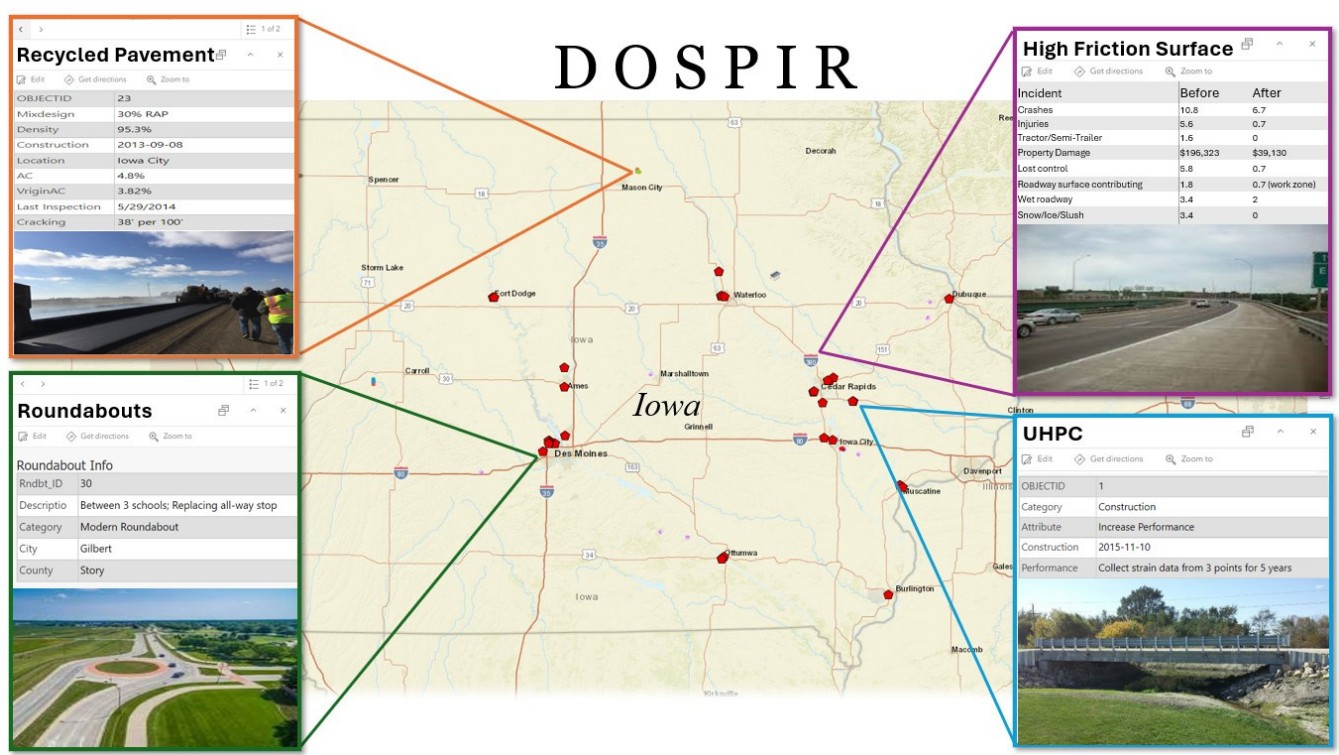

**Figure 3.** Screenshot of DOSPIR on ArcGIS Pro.

### 5.1. Field Conditions of High RAP Test Sections with Different Rejuvenators

Several high RAP test sections with two different rejuvenators for asphalt mixtures with three different RAP contents have been constructed in Cerro Gordo County, IA, USA. As illustrated in Figure 4, if a user clicks on this test section, DOSPIR will provide information about construction materials, asphalt mat densities, and cracking survey data of three different RAP contents with three different dosages of INVIGOR8 or TUFFTREK rejuvenators [18]. As can be seen from the blue box in the upper left corner of Figure 4, the field asphalt mat density of a test section of 34% RAP mix with PG58-28S and 3% TUFFTREK constructed on 3 August 2020, was 93.7%.

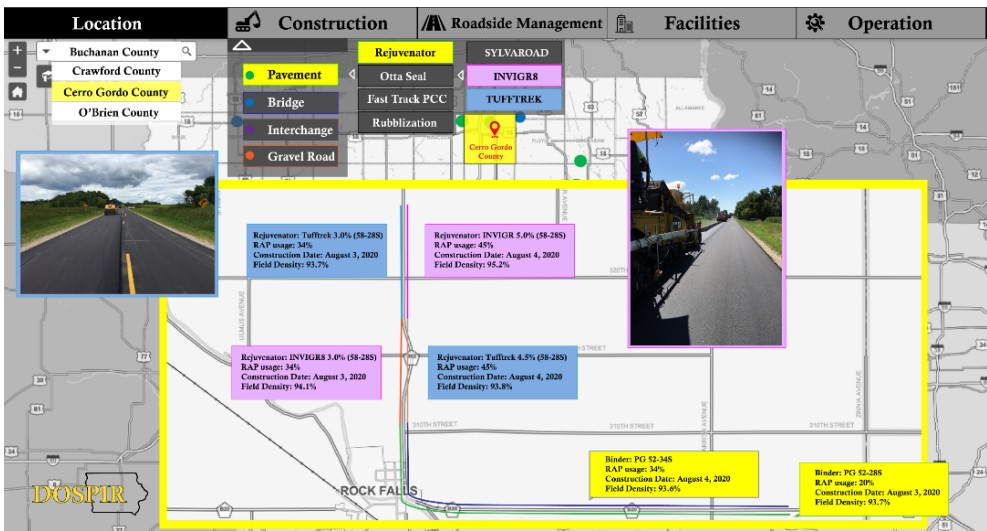

**Figure 4.** Illustration of asphalt pavement recycling and performance database.

*5.2. Characteristics and Performance of High-Friction Surface Treatments*

During the past decade, high-friction surface treatments (HFST) have been applied in nine sections in Iowa. As shown in Figure 5, each HFST section contains a picture of the test section showing crash locations along with the number of lanes, lane width, applied length, area, AADT, curve length, curve radius, runout length, and surface type. DOSPIR can be used to identify crash locations before and after HFST applications, which can be used to determine the effectiveness of HFST.

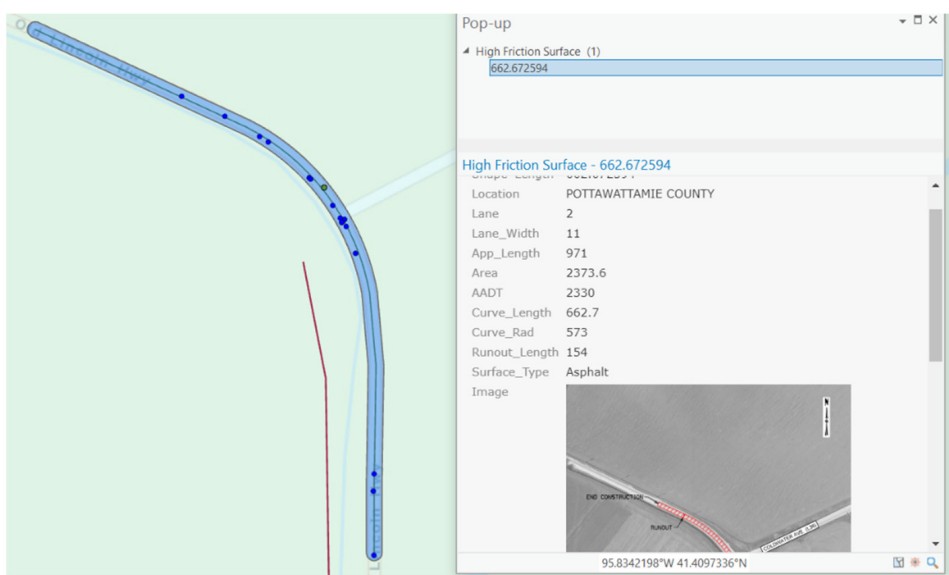

**Figure 5.** Illustration of high-friction surface treatment and crash database.

*5.3. Strain Database of a Bridge Built Using Ultra-High-Performance Concrete*

Recently, a new bridge was constructed using ultra-high-performance concrete (UHPC) in Buchanan County, Iowa [19]. As shown in Figure 6, if a user clicks on the menu item of UHPC, DOSPIR provides strain data from gauges installed on both sides of three joints over the past five years, along with a picture of a UHPC bridge. As can be seen from plots of strain data on the left (in dark blue color) and right (in light blue color) of each joint depicted at the bottom of Figure 6, it can be observed that strain values from both sides of all three joints are very close, which indicates good joint performance.

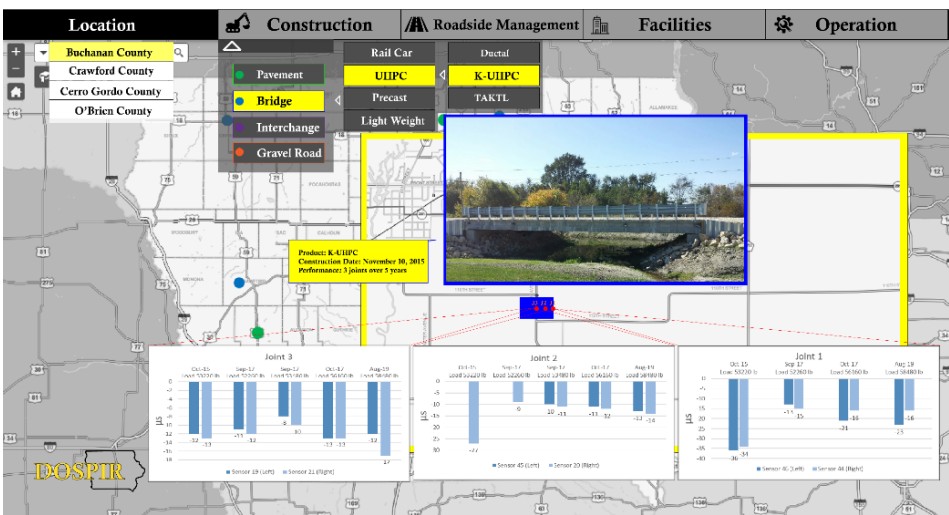

**Figure 6.** Illustration of UHPC bridge and performance database.

### 5.4. Effectiveness of Roundabouts in Crash Prevention

Over the past decade, a total of 110 roundabouts have been constructed in Iowa. This section discusses the effectiveness of roundabouts through both spatial and temporal analyses of traffic crash data near roundabouts. As shown in the left side of Figure 7, a user can draw a circle with a radius of 250 feet around the center of a roundabout and collect crash data. This feature merges two datasets about locations of roundabouts and crash sites by geographically aligning car crash data with nearby roundabout locations, specifically focusing on crashes that occurred within a 250-foot radius of each roundabout. This approach ensures that only crash data relevant to roundabouts are retained for analysis. To determine the impact of a roundabout on traffic crashes, the spatially joined data was then divided into two categories: "before" and "after" its construction. As shown on the right side of Figure 7, the number of crashes, damage level (1, low; 5, high), severity level (1, high; 5, low), and monetary value before and after the construction of a particular roundabout are illustrated. For this particular roundabout, the number of crashes slightly increased, but crash severity and damages decreased.

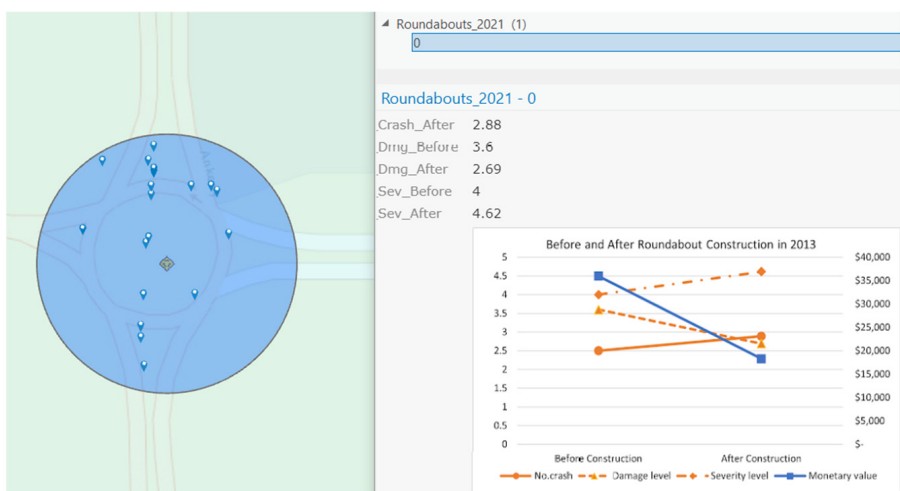

**Figure 7.** Illustration of roundabout and crash data analysis results.

### 6. Summary and Conclusions

The transportation sector contributes a significant amount of GHG emissions. Therefore, sustainable practices that reduce the negative environmental impacts of its operations

should be promoted. Sustainable transportation practices can yield (1) cost savings, (2) increased resource efficiency, (3) reduced environmental impacts, and (4) increased service life. Recently, to coordinate sustainability efforts effectively within Iowa DOT, the sustainability working group was established.

Based on the survey by AASHTO, less than half of state DOTs have adopted a definition of "sustainability" or considered sustainability as a factor in their strategic and project planning. Based on the survey by the University of Iowa, with a limited survey response, the top sustainability goal is "reuse and recycling." The most popular means to incorporate sustainability into the decision-making process is to modify specifications, which encourage more sustainable practices. Few state DOTs have sustainability evaluation metrics, but no state DOT provides any incentive for the implementation of sustainable practices.

There are many test sections constructed with various sustainable practices, and the evaluation results are commonly published in the report without saving the evaluation data in the online database. By analyzing sustainable practices across various states, we developed a methodology to find the most appropriate sustainable practices for Iowa's transportation system while improving environmental conditions and reducing the life-cycle cost. This paper presents efforts to develop a methodology for identifying the best sustainable practices for implementation in transportation infrastructure practices in Iowa. DOSPIR, a GIS database where construction, materials, and performance data of sustainable practices can be stored, was developed to help transportation engineers easily access past construction and performance data to identify the most appropriate sustainable practices by measuring the outcome of each sustainable practice.

The presented system is different from many existing pavement or bridge management systems, which store performance data for each uniformly segmented road segment without consideration of specific test section boundaries. There is no current GIS system with the capability of storing and analyzing construction and performance data collected from the test sections built using sustainable materials and practices.

DOSPIR is a Python program developed on ArcGIS Pro with a database of sustainable practices, which would allow users to access implementation records easily rather than go through the published reports. DOSPIR serves as a centralized repository of data on sustainable practices, which is a valuable tool for evaluating each sustainable practice by measuring its performance.

In conclusion, the goals of sustainable transportation infrastructures should be established based on objective performance data stored in the DOSPIR. This would significantly enhance the ability to monitor, assess, and predict the long-term sustainability impacts of transportation projects. The insights gained from this study will provide a roadmap for public transportation agencies to effectively implement sustainable practices.

**Author Contributions:** Conceptualization and methodology, H.L.; validation, B.M. and J.L.; investigation, B.M.; writing—original draft preparation, H.L. and J.L.; writing—review and editing, B.M. All authors have read and agreed to the published version of the manuscript.

**Funding:** This research was funded by the Iowa Department of Transportation (DOT), grant number IHRB Project SPR-398.

**Data Availability Statement:** The data presented in this study are available on request from the corresponding author.

**Acknowledgments:** The authors would like to thank the Technical Advisory Committee members for their guidance throughout this research project and state DOT employees who replied to the sustainability survey.

**Conflicts of Interest:** Byungkyu Moon was employed by the company Applied Research Associates, Inc. The remaining authors declare that the research was conducted in the absence of any commercial or financial relationships that could be construed as a potential conflict of interest. The funders had no role in the design of the study, in the collection, analyses, or interpretation of data, in the writing of the manuscript, or in the decision to publish the results.

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
