# Peer review of "Sustainable Transportation Infrastructures in Iowa—Goals and Practices"

_infrastructures, doi:10.3390/infrastructures9020027_

Round 1

Reviewer 1 Report

Comments and Suggestions for Authors

Infrastructures 2790542 review comments

This manuscript introduces the efforts conducted in the Iowa state to develop a database to document sustainability practices for further analysis and implementation. The research goals are effective and the results are of value to others of interest. The manuscript structure is okay but there are some minor grammar errors in it.

Title:

the authors may include “Iowa” in the title to emphasize that the work and product was developed for the state of Iowa.

Abstract:

The abstract is fine.

Introduction:

It is fine.

Background:

It is fine.

Sustainability Survey:

In the first paragraph of Section 3.1, please add citations to the references for the AASHTO SWG survey.

Sustainable Practices:

It is fine.

Database of Sustainable Practices…:

In Section 5, please describe what sustainable practices are included in DOSPIR.

What are the meanings of the symbols in the columns of Cost and Perf. in Table 2?

In Section 5.1, only density data are included in the database for High RAP test sections? How about other relevant data that are needed to analyze the performance of the test sections?

It is recommended to replace the word “accident” with “crash”.

In Line 268 and Line 270, ft is different from feet. Please use one expression for the unit throughout the manuscript. Also please note a typo (“feed”) in Line 314.

In Line 310, “were” should be “was”.

Spatial and Temporal Analyses…:

In this section, too many details (e.g., function or command names in Python or ArcGIS) on the procedure of database development are included, which are either unnecessary for readers who are familiar with Python or ArcGIS or useless for readers who do not have access to DOSPIR. It is recommended that more technical details be included, such as what measures are taken to ensure the data quality in DOSPIR, how the data in DOSPIR can be updated with new inputs and at what frequencies.

Summary and Conclusions:

The acronym DOSPIR in line 368 has been defined previously.

Comments on the Quality of English Language

There are some minor grammar errors or typos.

Reviewer 2 Report

Comments and Suggestions for Authors

The subject of the article is interesting and the problem is worth exploring.
It would be necessary to characterize the authors' barriers and limitations in their experiments.

The article does not contain much scientific content, while its practical and overview value is an undoubted advantage. The authors pointed out the possibility of using GIS programming for infrastructure management. Supplementing the paper with an additional "Discussion of Results" section would be reasonable. In discussing the results, it would be appropriate to refer to other studies in this area and point out the differences, advantages and disadvantages of the solutions presented by the authors and indicate the contribution to scientific research. In addition to showing the possibilities of analysis using DOSPIR, it would be reasonable to comment on the results.

Reviewer 3 Report

Comments and Suggestions for Authors

The first half of this paper is interesting and runs without major problems. However, the second half (section 6 on) is problematic and makes the whole paper not appropriate for publication. Here are some detailed remarks that will explain the above overall judgment:

 L50, do you mean “systemic methodology” or “systematic”? (you mention this also in L360). If you mean “systemic” I don’t understand what is “systemic” about the methodology you use.

L55, cross out the phrase “This paper” as you have used it in the previous sentence.

L108, put parenthesis after the word “water” i.e. “water),”

L218, Figure 3 is not legible.

L225-227, Since Figure 3 is not legible by the reader, it is suggested to enhance the explanations given about this Figure in these two lines. You must expand the sentence: “As can be seen from 226 Figure 3, field asphalt mat density of PG58-28S with 45% RAP and TUFFTREK was 93.8%.”.

L261, Put the phrase “special join” in “”. The same in L283.

L250 – 351, The whole of section no. 6, is confusing and needs to be re-written. Here are some of the findings that make me say so:

a.      Sections 6.1, 6.2, and part of 6.3 until L309, are written for computer programmers and do not convey any meaning or value to the non-expert programmer reader. It is suggested that they be removed.

b.     There are two Figure 4s.

c.      Figures 4 (both) and Figure 10 (probably it is meant as Figure 6) are not legible and should be removed. In any case, they do not offer any utilitarian info to the reader.

d.     In L295, Figure 5 is mentioned but there is no such Figure in the text. After the two Figure 4s, there is a Figure numbered 10!

e.     The same (i.e. they do not appear in the text) applies to Figure 6 (L310), and Figure 7 (L315),

f.       After L323, the authors should expand the text by referring to a) the accuracy and reliability of these results, and b) whether the mentioned differences are statistically significant or not. This is necessary because the values of the “Before” and the “After” situations seem very close to each other and the reader needs to know if these differences represent real changes. This goes especially for the property damage results (L322) which are subject to subjective estimates anyway.

g.     The severity of accidents needs to be mentioned and discussed in more detail and separately from the property damage as it is now. First, you should define “severity”, i.e., how do you define and measure it? What do the figures in the vertical axis of Figure 12 represent, and so on?

L352, the conclusions section is also problematic. The conclusions mentioned there, are not reflected in the analyses and findings of the previous texts. The creation of the DOSPIR database is a valuable result reported in the paper but its use to derive results concerning the efficiency of traffic operations, the safety of road usage, or the change in decision-making processes, is not well documented and remains unproven.

Round 2

Reviewer 2 Report

Comments and Suggestions for Authors

I accept the article in its present form.

Reviewer 3 Report

Comments and Suggestions for Authors

No more comments. Thank you for your revisions.